# Longitudinal Motor Functional Outcomes and Magnetic Resonance Imaging Patterns of Muscle Involvement in Upper Limbs in Duchenne Muscular Dystrophy

**DOI:** 10.3390/medicina57111267

**Published:** 2021-11-18

**Authors:** Claudia Brogna, Lara Cristiano, Tommaso Verdolotti, Giulia Norcia, Luana Ficociello, Roberta Ruiz, Giorgia Coratti, Lavinia Fanelli, Nicola Forcina, Giorgia Petracca, Fabrizia Chieppa, Tommaso Tartaglione, Cesare Colosimo, Marika Pane, Eugenio Mercuri

**Affiliations:** 1Pediatric Neurology Unit, Fondazione Policlinico Universitario “A. Gemelli”, IRCCS, 00168 Rome, Italy; claudiabrogna@yahoo.it; 2Nemo Clinical Centre, Fondazione Policlinico Universitario “A. Gemelli”, IRCCS, 00168 Rome, Italy; giulianorcia@yahoo.it (G.N.); giorgia.coratti@unicatt.it (G.C.); lavinia.fanelli@gmail.com (L.F.); n.forcina@gmail.com (N.F.); marika.pane@policlinicogemelli.it (M.P.); 3Pediatric Neurology, Università Cattolica del Sacro Cuore, 00168 Rome, Italy; laralu84@hotmail.it (L.C.); robertaruiz1994@gmail.com (R.R.); giorgia.petracca@libero.it (G.P.); fabrichieppa@gmail.com (F.C.); 4Institute of Radiology, Fondazione Policlinico Universitario “A. Gemelli”, IRCCS, 00168 Rome, Italy; tommaso.verdolotti@policlinicogemelli.it (T.V.); luana.ficociello@policlinicogemelli.it (L.F.); cesare.colosimo@policlinicogemelli.it (C.C.); 5Department of Radiology, Istituto Dermatologico Italiano, IRCCS, 00167 Rome, Italy; tommaso.tartaglione@unicatt.it; 6Institute of Radiology, Università Cattolica del Sacro Cuore, 00168 Rome, Italy

**Keywords:** muscle MRI, PUL, DMD

## Abstract

*Background and Objectives*: The aim of this study was to evaluate longitudinal changes using both upper limb muscle Magnetic Resonance Imaging (MRI) at shoulder, arm and forearm levels and Performance of upper limb (PUL) in ambulant and non-ambulant Duchenne Muscular Dystrophy (DMD) patients. We also wished to define whether baseline muscle MRI could help to predict functional changes after one year. *Materials and Methods*: Twenty-seven patients had both baseline and 12month muscle MRI and PUL assessments one year later. *Results*: Ten were ambulant (age range 5–16 years), and 17 non ambulant (age range 10–30 years). Increased abnormalities equal or more than 1.5 point on muscle MRI at follow up were found on all domains: at shoulder level 12/27 patients (44%), at arm level 4/27 (15%) and at forearm level 6/27 (22%). Lower follow up PUL score were found in 8/27 patients (30%) at shoulder level, in 9/27 patients (33%) at mid-level whereas no functional changes were found at distal level. There was no constant association between baseline MRI scores and follow up PUL scores at arm and forearm levels but at shoulder level patients with moderate impairment on the baseline MRI scores between 16 and 34 had the highest risk of decreased function on PUL over a year. *Conclusions*: Our results confirmed that the integrated use of functional scales and imaging can help to monitor functional and MRI changes over time.

## 1. Introduction

In the last few years, the development of new outcome measures has allowed a better definition of the natural history of Duchenne Muscular Dystrophy (DMD), a progressive, X-linked neuromuscular disorder caused by mutation in the dystrophin gene, affecting one in 3600–5000 live male births. The absence of functional dystrophin protein leads to a progressive muscle degeneration, weakness and to a predictable pattern of loss of specific functional milestones [1,2,3,4,5,6,7,8,9]. 

While in the past the severity of muscle involvement was mainly assessed by muscle biopsy, over the last two decades there has been increasing attention to the use of non-invasive techniques, such as Magnetic Resonance Imaging (MRI) and spectroscopy (MRS) as markers of muscle pathology and disease progression [10,11,12,13,14,15,16,17,18,19,20]. Most MRI studies have focused on lower limbs and less has been reported on possible changes in upper limb involvement. Furthermore, most of the available studies have focused on the assessment of a portion of upper limbs, either distal or proximal [21,22,23,24,25]. As suggested in our recent cross-sectional study investigating all upper limb domains (proximal, mid, distal) in combination with functional scale including Performance of Upper Limb (PUL) test [26] a wider approach may help to better understand the progression of upper limb muscle involvement over time. Despite very well known clinical evidence of proximal to distal clinical progression of upper limb involvement, the changes in the different domains do not occur following a well defined linear progression as early signs of distal involvement can already be found in relatively young ambulant DMD boys who still have some preserved shoulder function [27]. 

The aim of the present study was to evaluate longitudinal changes using both upper limb muscle MRI at shoulder, arm and forearm levels and PUL in ambulant and non-ambulant DMD patients. Furthermore, following evidence from lower limb MRI studies showing that subtle muscle MR changes may precede functional changes in boys with DMD [28,29] we also wished to define whether baseline muscle MRI could help to predict functional changes in the upper limbs. 

## 2. Methods

The study is part of a project aimed at establishing muscle MRI changes in DMD. The study was approved by the Ethics committee of our Institution (Protocol ID code numbers 0014186/17, approved on 21 March 2017). Written informed consent was obtained for all the patients who agreed to participate. For the minors, consent was obtained by their parents. As part of this study, we enrolled DMD patients attending their routine follow up clinics between June 2017 and December 2018. The exclusion criteria were related to the impossibility to perform MRI without sedation, therefore excluding very young children or those with severe cognitive or behavioral problems. We also excluded patients with severe joint contractures, pacemakers, respiratory or cardiac problems that would interfere with positioning or performing MRI. The investigations were carried out following the rules of the revised Declaration of Helsinki.

### 2.1. Muscle MRI

Unilateral upper-limb MRI was performed at 1.5T (Philips Ingenia, Philips Healthcare, Best, The Netherlands) using a flexible body coil (body SENSE 32 Channel Flex Coil). Subjects lay in the scanner in the head-first supine position, with the dominant upper limb to be imaged lying in a comfortable position on the scanner bed alongside the torso. The upper limb was stabilized using a fabricated thermoplastic splint and sandbags were placed over the forearm and hand to minimize motion. Boys were encouraged to watch a movie or cartoon during scan acquisition and usually a parent and/or a staff member was present in the scan room during scan acquisition. The patients did not receive any sedation and the total examination time was approximately 20 to 30 min.

Non contrast-enhanced images were obtained from the dominant upper limb.

TSE T1-weighted spin-echo was acquired on axial plane selected in respect to the long axis of the humerus for the shoulder and arm, and in respect of the long axis of the radius for the forearm. The slices were set up to cover the entire extension of the upper limb. Each muscle was evaluated throughout its length.

Scan parameters were as follows: TSE T1 (FOV 160 × 160 mm, voxel 0.8 × 0.8, matrix 200 × 200, 5 mm axial slices, slice gap 0 mm, flip angle 90°, TR 150ms, TE 8 ms).

Descriptive analysis was used to identify the muscles that were more frequently affected in the different segments. At shoulder level the muscles examined were: deltoid, supraspinatus, infraspinatus, subscapularis, coraco-brachialis, pectoralis major and minor, teres minor, latissimus dorsi, serratus anterior. At arm level the muscles examined were: biceps brachii, brachialis and triceps brachii. At the forearm level the muscles examined were: supinator, pronator teres, flexor carpi radialis, palmar, flexor digitorum superficialis, flexor carpi ulnaris, flexor digitorum profundus, anconeus, flexor pollicis longus, extensor carpi ulnaris, extensor digiti minimi, extensor digitorum, extensor carpi radialis, brachioradialis, extensor pollicis longus.

For each segment, we identified the muscles that were more often spared or affected. Although in the present paper we did not aim to quantify the level of involvement, as this will be separately reported, we used a previously reported classification to have a rough estimate of the level of involvement. All muscles MRI scans were assessed for normal or abnormal signal intensity within the different muscles groups and scored using Mercuri classification [16,17], as follows: Stage 0: normal appearance; Stage 1: scattered small areas of increased intensity on T1W images; Stage 2a: numerous discrete areas of increased intensity on T1W images involving less than 30% of the volume of the muscle; Stage 2b: numerous discrete areas of increased intensity on T1W images with early confluence of, 30–60% of the volume of the muscle; Stage 3: washed-out appearance due to confluent areas of increased intensity on T1W images with muscle still present at the periphery; Stage 4: end-stage appearance, muscle entirely replaced by areas of increased intensity on T1W images. We arbitrarily subdivided muscles with normal MRI or with only minimal changes (score 0 and 1), those with intermediate involvement (grades 2 to 3) and those with complete replacement (grade 4). We also subdivided MRI score at baseline in 3 group for each segment of upper limb (shoulder, arm, forearm) according to different progressive grading of muscle involvement at follow up (mild-moderate-severe). For each segment the sum of the scores of all the muscles analysed has been considered: at shoulder level total shoulder scores were considered as mild if <16, moderate if between 16–34 and severe when >34. At arm level total arm scores were considered as mild if <4, moderate if between 4–7.5, severe when >7.5. At forearm total forearm scores were considered as mild if ≤6, moderate if between 6.5–36, severe when >36. 

Three examiners (LC, TT and TV) scored the scans separately with consensus on the scores of over 90%, In the discordant cases the scans were reviewed by all examiners and an agreement was found.

### 2.2. PUL 2.0

The PUL 2.0 (Appendix A) includes an entry item to define the starting functional level, and 22 items subdivided into shoulder level (6 items), middle level (9 items) and distal level (7 items) dimension [7,8]. The entry items are based on a revised version of the Brooke score and range from score 0–no useful hand function-to score 6 full shoulder abduction–no weakness. For weaker patients a low score on the entry item means high level items do not need to be performed. Each dimension (shoulder, middle, distal) can be scored separately according to a score defined (ex. 2: able; 1: with compensation; 0: unable). The 22 items includes item 1: shoulder abduction (both arms above head); item 2: shoulder abduction to shoulder height; items 3 (no weights) and 4 (500 g): shoulder flexion to shoulder height; items 5 (500 g) and 6 (1 kg): shoulder flexion above shoulder height; item 7: hands to mouth; item 8: hands lap to table; items 9, 10, and 11: move weight on table (100 g, 500 g, 1 kg); item 12: lift heavy can diagonally-timed; item 13 (three cans) and 14 (five cans). No time; item 15: remove lid; item 16: tearing paper; item 17: tracing path; item 18: push on light; item 19: supination; item 20: pick up coins; item 21: number diagram; item 22: pick up 10 g finger pinch. 

In the PUL 2.0 there is a maximum score of 12 for the shoulder level, 17 for the middle level, and 13 for the distal level. A total score can be achieved by adding the three level scores (max global score 42). 

### 2.3. Baseline MRI and Follow up PUL 

In order to predict functional changes we reported MRI scores at baseline and PUL scores at follow-up.

### 2.4. Statistical Analysis

Descriptive statistics (N, mean, median, SD, Range) were used. T student was used to compare differences in total scores and subtotal scores of each level (shoulder, arm and forearm) between baseline and follow up for both MRI and PUL. The correlation between MRI scores at baseline and PUL scores at follow up for each level was explored with Pearson correlation test. The level of significance was set at *p* < 0.05.

## 3. Results

Thirty-one patients who had muscle MRI at baseline were enrolled in the study. Twenty-seven of the 31 patients completed the study one year later. Three of the remaining 4 were enrolled in clinical trials and the other one refused to repeat the scan. Ten of the 27 patients were ambulant (age range 5–16 years), and 17 non ambulant (age range 10–30 years). All the patients were on steroids.

No significant differences were found at baseline and at follow up for both MRI and PUL in total scores and subtotal scores at each level (shoulder, arm and forearm). No significant correlation was found between baseline MRI total scores, baseline MRI subtotal scores of each level and PUL scores at follow up but for the shoulder level where a moderate correlation (r = 0.41, *p* = 0.01) was found between MRI shoulder scores at baseline (between 16 and 34), and PUL shoulder scores at follow up.

### 3.1. Total Scores

#### 3.1.1. Muscle MRI

None of the patients had a completely normal MRI at baseline or at follow up. 

The total scores at baseline ranged between 12 and 110 (mean 45.98, SD ± 29.69); the total score at follow up ranged between 12.5 and 110 (mean 49.85, SD ± 30.22); MRI changes ranged between 0 and + 14 (mean 3.87, SD ± 4.20) (For details Figure 1 and Figure 2 (Appendix A)). 

#### 3.1.2. PUL 2.0

The total scores at baseline ranged between 8 and 42 (mean 33.74, SD ± 10.64); the total score at follow up (1 year later) ranged between 6 and 42 (mean 30.59, SD ± 10.98); PUL changes ranged between −9 and +2 (mean −2.14, SD ± 2.44) (For details Figure 2, Figure 3). 

### 3.2. Shoulder Level 

#### 3.2.1. Muscle MR at Baseline and at Follow up 

At shoulder level 11/27 (41%) patients had stable scores between baseline and 1 year (changes = 0/0.5), 4/27(15%) + 1 point and 12/27 (44%) showed changes equal or more than 1.5. (For details Figure 4 and Figure 5). 

Latissimus dorsi and serratus anterior were the most often severely involved muscles both at baseline and at follow up, followed by the subscapularis, infraspinatus and the sopraspinatus with a score of 2 or higher in 27/27 (100%) of patients. Ten of 27 patients (37%) showed a worsening (a higher MRI score than baseline) in at least 2 or more than 2 muscles. The changes were most frequent in the deltoid muscle (10/27) (37%), followed by latissimus dorsi (8/27) (30%). Pectoralis major and minor and coracobrachialis were the muscles that showed less often changes at follow up. 

Of the 10 patients with baseline MRI shoulder scores lower than 16, five (50%) had higher scores (>1.5) on follow up MRI.

Of the 11 patients with baseline MRI scores between 16 and 34, 6 (54%) patients had higher shoulder scores (>1.5) on follow up MRI.

Of the 6 patients with baseline MRI scores >34, 1 (17%) had higher scores (>1.5) on follow up MRI.

#### 3.2.2. PUL

At shoulder level 19/27 (70%) patients had stable scores between baseline and 1 year (changes = 0/± 1); 5/27 (18%) showed changes between −2 and −3, and 3/27 (11%) showed changes between −4 and −5 (For details Figure 5 and Figure 6)

#### 3.2.3. MRI at Baseline and Follow up PUL

Ten patients had baseline shoulder MRI scores lower than 16: 9 of the 10 (90%) showed stable and 1 (10%) lower PUL scores.

Eleven patients had baseline MRI scores between 16 and 34: 4 of the 11 (36%) had stable and 7 (64%) lower PUL scores.

Six patients had baseline MRI scores >34: all 6 had a baseline PUL score of 0 with no possibility to show further loss (For details Figure 5).

### 3.3. Arm Level

#### 3.3.1. Muscle MR at Baseline and at Follow up

At arm level 17/27 (63%) patients had stable scores between baseline and 1 year (changes = 0), 6/27 (22%) +1 point, 4/27 (15%) showed changes equal or >1.5. (For details Figure 7 and Figure 8).

Most patients showed a concordant involvement of all the three muscles (biceps, brachialis and triceps) with a similar proportion of patients with a score of 2 or higher with the brachialis showing the highest number of scores >3.

Of the 11 patients with baseline MRI scores equal or lower than 4, 9 (82%) had stable scores on follow up MRI, 2 (18%) had a score of +1 point.

Of the 10 patients with baseline MRI scores between 6 and 7.5, 4 of the 10 (40%) had higher scores (>1.5) on follow up MRI.

Of the 6 patients with baseline MRI scores >7.5, 5 (83%) had stable scores on follow up MRI, 1 (17%) had a score of +1 point.

#### 3.3.2. PUL

At Mid-level 17/27 (63%) patients had stable scores between baseline and 1 year (changes = 0/±1), 8/27 (30%) showed changes between −2 and −3, and 1/27 (4%) with changes below 4 (For details Figure 8 and Figure 9). One patient (4%) showed an improvement of 2 points.

#### 3.3.3. MRI at Baseline and Follow up PUL

Eleven patients had baseline arm MRI scores lower than 4: 10 of the 11 (91%) had stable and 1 (9%) lower PUL scores.

Ten patients had baseline MRI scores between 6 and 7.5: 4 of the 10 (40%) had stable and 6 (54%) lower PUL scores.

Six patients had baseline MRI scores >7.5: 3 of the 6 (50%) showed stable, 1 (17%) had + 2 points, and 2 (33%) lower PUL scores (For details, Figure 8).

### 3.4. Forearm Level

#### 3.4.1. Muscle MR at Baseline and at Follow up

At foream level 20/27 (74%) patients had stable scores between baseline and 1 year (changes = 0), 1/27 (4%) + 1 point, 6/27 (22%) showed a changes score above 1.5. (For details Figure 10 and Figure 11).

At foream level supinator and pronator were the most frequently involved muscles on MRI both at baseline and at follow up with a score of 2 or higher in more than 75% of patients, followed by flexor carpi ulnaris (59%) and brachioradialis (52%).

Of the 11 patients with baseline MRI scores lower than 6 all (100%) had stable scores on follow up MRI.

Of the 11 patients with baseline MRI scores between 6.5 and 36, 4 (36%) had higher scores on follow up MRI.

Of the 5 patients with baseline MRI scores more than 36, 2 (40%) had higher scores on follow up MRI (2/5 in supinator and extensors muscles), 1 (2%) had a score of +1.

#### 3.4.2. PUL

At distal level 26 of the 27 (96%) patients had stable scores at follow up (changes = 0/± 1), and the remaining one (4%) had changes above 2. (For details Figure 11 and Figure 12).

#### 3.4.3. Muscle MRI at Baseline and Follow up PUL

All patients had PUL stable scores at follow up irrespective of their MRI baseline score (For details, Figure 11).

## 4. Discussion

Our longitudinal study systematically assesses both muscle MRI and upper limb function at different levels, including shoulder, arm and forearm, in a cohort of ambulant and non-ambulant DMD patients. The use of longitudinal muscle MRI allowed to identify the patterns of progression at each level. The changes observed in our cohort are consistent with other studies also reporting longitudinal MRI and MRS findings focusing on one upper limb segment (mainly forearm) [21,22,23,24] on all upper limb domains [25].

The possibility to explore all the domains allowed to define which muscles and domains are at higher risk of becoming progressively involved with increased age. MRI changes of at least two points were found in 15/27 (55%) patients on at least one domain explored, with patients showing changes in two (*n* = 6), or three (*n* = 3) domains. MRI changes were more often found at shoulder domain, probably related to the higher number of relatively young ambulant patients in our cohort. Although the numbers were relatively small to allow a meaningful statistical analysis taking into account so many variables, it is of interest that in each domain the changes indicating a more severe impairment mainly occurred in muscles which had mild to moderate impairment (between 19 and 34) at baseline and more rarely occurred in the muscles who were more severely impaired or were completely spared at baseline.

The MRI scores at baseline could not always predict the risk of showing functional changes over one year. The PUL total scores showed a decrease in 15/27 (*n* = 55%) and an overall mean reduction of 2 points/year, in line with previous studies reporting 12 month changes in larger cohorts [7,27]. There was no constant association between baseline MRI scores and follow up PUL scores but at shoulder level the highest risk of decreased function on PUL over a year was associated with moderate impairment on the baseline MRI scores (between 16 and 34) (r = 0.41, *p* = 0.01). Smaller functional changes were found at the two ends of the MRI spectrum. In patients with low MRI scores at baseline the low risk of fucntional redution was probably related to the relatively preserved muscles. At the other end of the spectrum, in patients with very high MRI scores indicating severe muscle impairment, the PUL scores were already extremely low at baseline and could therefore not show much further deterioration.

At arm and forearm level MRI did not appear to be equally sensitive to detect PUL changes. These results should however be interpreted with caution because of the limited overall number of patients in our cohort and the relatively low number of non-ambulant patients that may have reduced the possibility to observe more changes in the mid or distal domain. Another limitation of the study is that we only performed a visual analysis and better correlation may have was related to the been obtained by using a quantitative MR approach.

Another limitation was that related to the exclusion criteria as MRI cannot be easily perfomed in patients with severe behavioural or cognitive problems or severe contractures.

## 5. Conclusions

Even with these limitations our results confirm that muscle MRI could represent a reliable biomarker to monitor upper limb muscle involvement also in young DMD patients [25,29]. Our results also suggest that the integrated use of imaging and functional scales may at least partially help to identify patients at higher risk of functional or MRI changes.

This information can be useful at the time of designing clinical trials or intervention studies to exclude or select patients or to select the most appropriate imaging protocol. The follow up of these patients and the results of the quantitative analysis, both in progress, are needed to provide more accurate information on longitudinal changes at all levels.

## Figures and Tables

**Figure 1 medicina-57-01267-f001:**
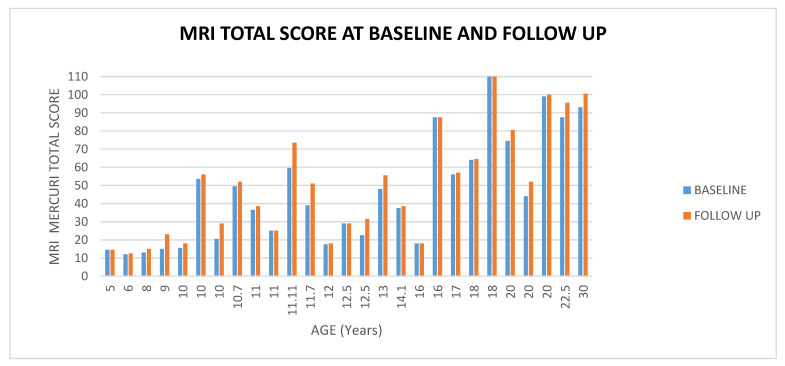
Showed muscle MRI total score at baseline and at follow up according to age.

**Figure 2 medicina-57-01267-f002:**
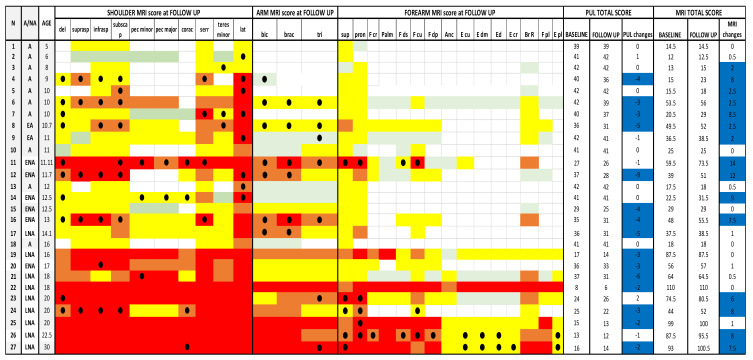
MRI and PUL scores at shoulder, arm and forearm level at baseline and at follow up. Individual details of imaging and PUL findings at shoulder, arm and forearm level. The shading reflects the severity of involvement with the score of 0 shown as a white cell, score of 1 as pale green, score of 2 as yellow, score of 3 as orange and score of 4 as red. Del = deltoid; suprasp = supraspinatus; infrasp = infraspinatus; subscap = subscapularis; pec = pectoralis; corac = coraco-brachialis; serr = serratus anterior; lat = latissimus dorsi; bic = biceps brachii; brac = brachialis; tri = triceps brachii; sup = supinator;pron = pronator teres; F cp = flexor carpi radialis; palm = palmar; F ds = flexor digitorum superficialis; F cu = flexor carpi ulnaris; Fdp = flexor digitorum profundus; Anc = anconeus; E cu = extensor carpi ulnaris; Edm = extensor digiti minimi; E d = extensor digitorum; E cr = extensor carpi radialis; Br R = brachioradialis; F pl = flexor pollicis longus; E pl = extensor pollicis longus. A = ambulant; EA = early ambulant; NA= non ambulant; ENA = early non ambulant; LNA: late non ambulant. ● Black circle means a higher Mercuri score respect to the baseline in the respective muscle. Blue line indicates changes in both MRI and PUL scores.

**Figure 3 medicina-57-01267-f003:**
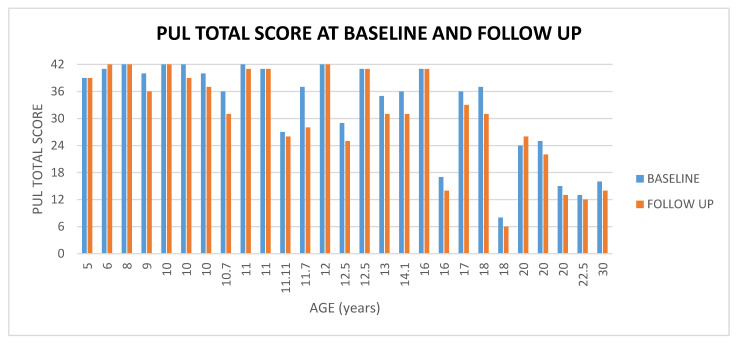
Showed PUL total score at baseline and at follow up according to age.

**Figure 4 medicina-57-01267-f004:**
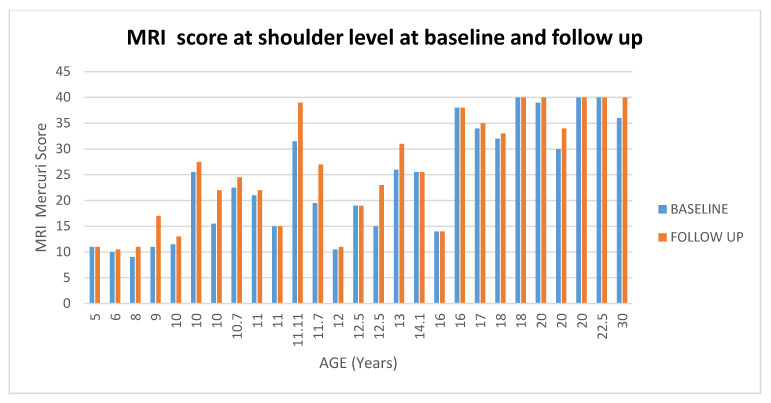
Showed scores in muscle MRI at shoulder level at baseline and at follow up according to the age.

**Figure 5 medicina-57-01267-f005:**
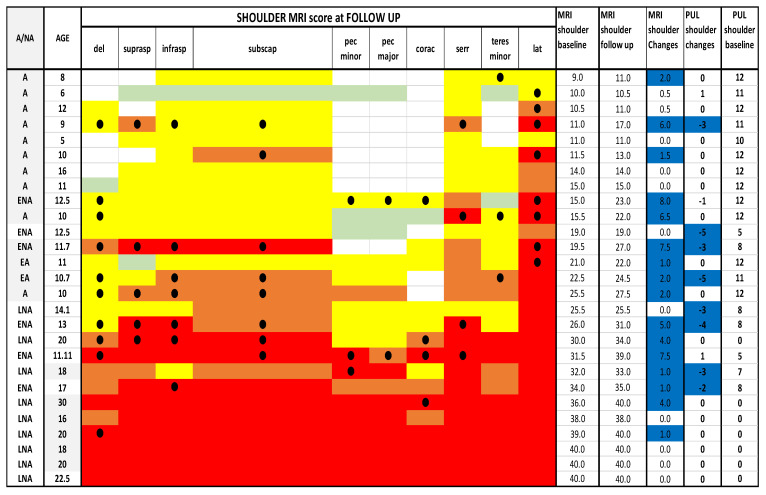
Detail of MRI and PUL at shoulder level. Individual details of imaging and PUL findings at shoulder level. The shading reflects the severity of involvement with the score of 0 shown as a white cell, score of 1 as pale green, score of 2 as yellow, score of 3 as orange and score of 4 as red. The grade 2.5 was used to identify patients with 2b involvement. Del = deltoid; suprasp = supraspinatus; infrasp = infraspinatus; subscap = subscapularis; pec = pectoralis; corac = coraco-brachialis; serr = serratus anterior; lat = latissimus dorsi; A = ambulant; EA = early ambulant; NA = non ambulant; ENA = early non ambulant; LNA: late non ambulant. ● Black circle means a higher Mercuri score respect to the baseline in the respective muscle. Blue line indicates changes in both MRI and PUL scores.

**Figure 6 medicina-57-01267-f006:**
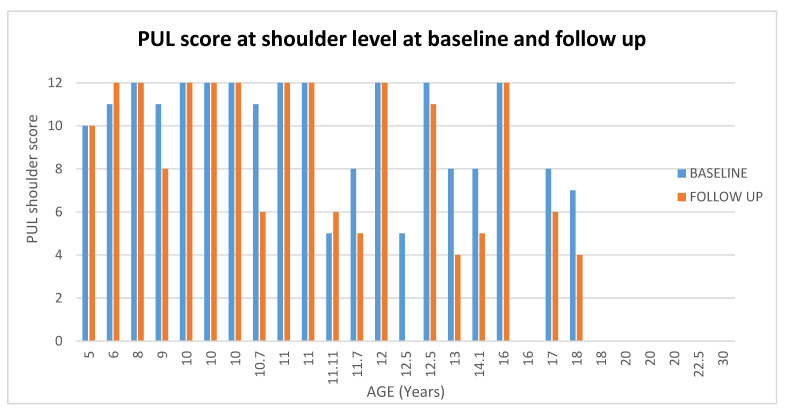
Showed scores in PUL at shoulder level at baseline and at follow up according to age.

**Figure 7 medicina-57-01267-f007:**
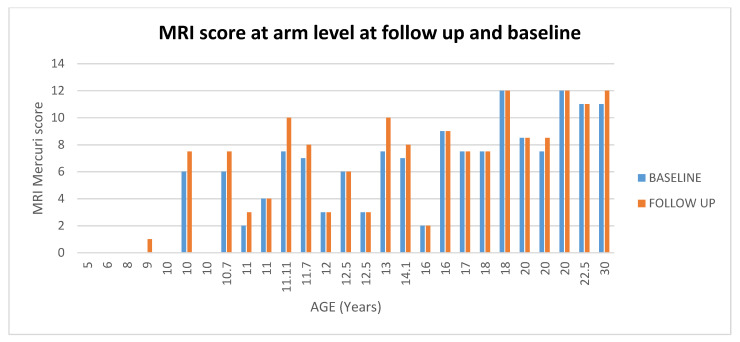
Showed scores in muscle MRI at arm level at baseline and at follow up according to age.

**Figure 8 medicina-57-01267-f008:**
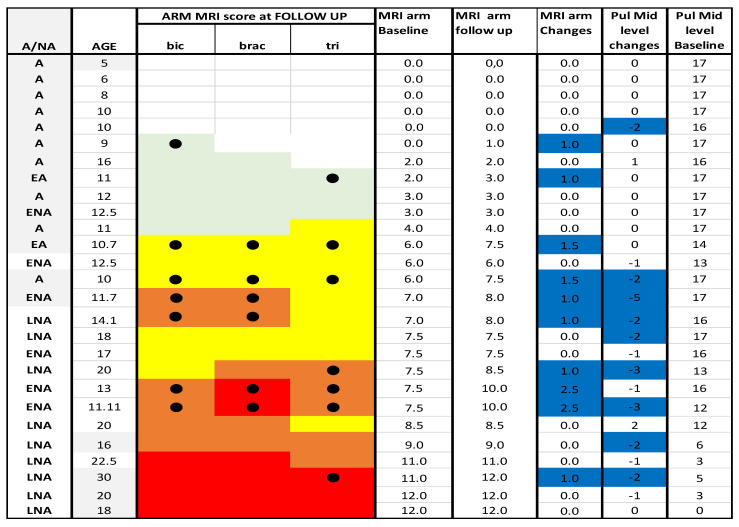
Detail of MRI and PUL at arm level. Individual details of imaging and PUL findings at arm level. The shading reflects the severity of involvement with the score of 0 shown as a white cell, score of 1 as pale green, score of 2 as yellow, score of 3 as orange and score of 4 as red. The grade 2.5 was used to identify patients with 2b involvement. Bic = biceps brachii; brac = brachialis;tri = triceps brachii; A = ambulant; EA = early ambulant; NA = non ambulant; ENA = early non ambulant; LNA: late non ambulant. ● Black circle means a higher Mercuri score respect to the baseline in the respective muscle. Blue line indicates changes in both MRI and PUL scores.

**Figure 9 medicina-57-01267-f009:**
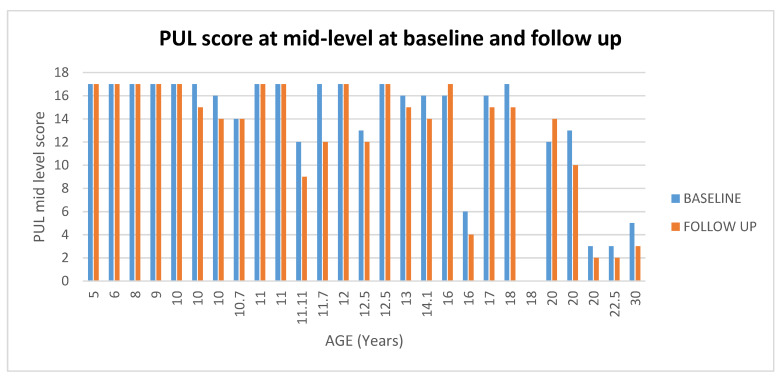
Showed scores in PUL at Mid-level at baseline and at follow up according to the age.

**Figure 10 medicina-57-01267-f010:**
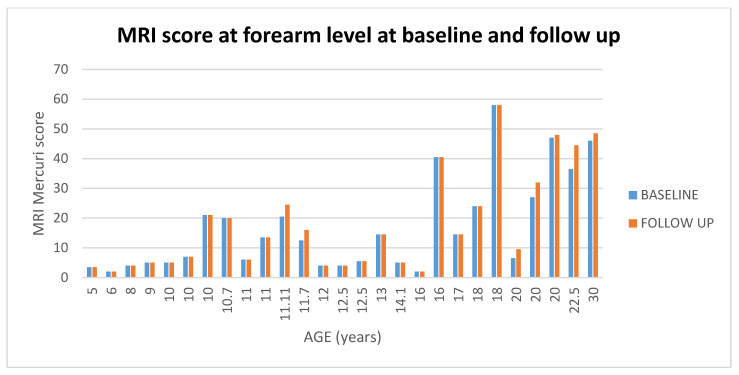
Showed scores in muscle MRI at distal level at baseline and at follow up according to age.

**Figure 11 medicina-57-01267-f011:**
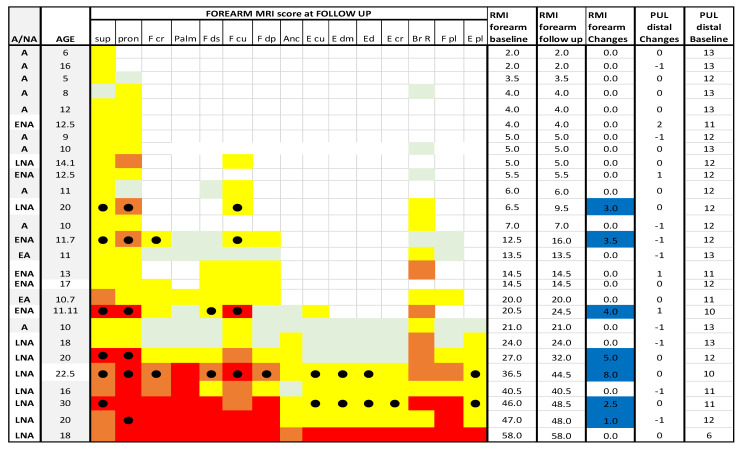
Detail of MRI and PUL at forearm level. Individual details of imaging and PUL findings at forearm level. The shading reflects the severity of involvement with the score of 0 shown as a white cell, score of 1 as pale green, score of 2 as yellow, score of 3 as orange and score of 4 as red. The grade 2.5 was used to identify patients with 2b involvement. Sup = supinator; pron = pronator teres; F cp = flexor carpi radialis; palm = palmar; F ds = flexor digitorum superficialis; F cu = flexor carpi ulnaris; Fdp = flexor digitorum profundus; Anc = anconeus; E cu = extensor carpi ulnaris; Edm = extensor digiti minimi; E d = extensor digitorum; E cr = extensor carpi radialis; Br R = brachioradialis; F pl = flexor pollicis longus; E pl = extensor pollicis longus. A = ambulant; EA = early ambulant; NA = non ambulant; ENA = early non ambulant; LNA: late non ambulant. ● Black circle means a higher Mercuri score respect to the baseline in the respective muscle. Blue line indicates changes in MRI scores.

**Figure 12 medicina-57-01267-f012:**
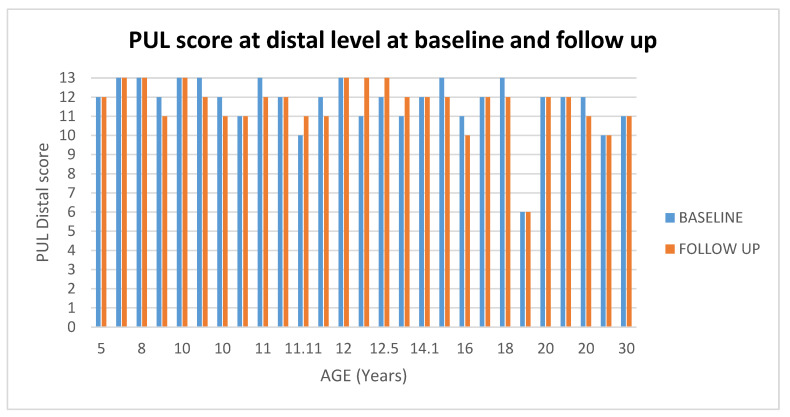
Showed scores in PUL at distal level at baseline and at follow up according to age.

## Data Availability

Data is contained within the article or supplementary material.

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
