# Peer review of "Longitudinal Motor Functional Outcomes and Magnetic Resonance Imaging Patterns of Muscle Involvement in Upper Limbs in Duchenne Muscular Dystrophy"

_medicina, 2021, doi:10.3390/medicina57111267_

Round 1
Reviewer 1 Report
The present work investigates the MRI and PUL variations over 1 year of observation in a sample of 27 DMD patients, in order to assess the feasibility and reliability of an integrated muscle imaging and functional testing in monitoring disease output and progression. The study is very interesting, as the relevance of the findings could be of great help in the clinical routine and practice when treating DMD patients; thus, I thank the author for their efforts.
However, there are few points which I believe could be addressed in order to strengthen the results of this work, and also some suggestions to potentially help the paper becoming more reader-friendly.
- Methods: exclusion criteria. I understand you had to exclude those patients where MRI could not be collected for many reasons (cited in the paper). However, I was wondering whether this could represent a limitation of this work, as the proposed method for measuring muscle MRI and PUL cannot be applied to all patients. I understand there is now way to solve this, but the authors could cite it as a limitation of the study.
- Methods: MRI scoring, L110-116: I don’t understand how you defined the score. Was it the sum of the scores of all the muscles analysed for each district (shoulder, arm and forearm)? I.e. Did you sum the individual score of, for example, biceps brachii, brachialis and triceps brachii to obtain the score of the arm?
L199: PUL: I see there are references, however I would require the authors to briefly explain (in a more comprehensive manner) the PUL determination modality. When I read the paper indicated as ref. 8, it becomes clearer what the PUL 2.0 is; of course the authors cannot report the whole questionnaire, but rephrasing the chapter in a clearer way would certainly help an inexpert reader. For example, briefly explaining what the entry item is, and providing few examples of the other items (i.e. hand to mouth, hands lap to table) and how the score is defined (ex. 2: able; 1: with compensation; 0: unable). This would be particularly useful also at the light of the fact that all the results are presented referring to the increased/decreased PUL scores, and the reader has no idea of the functional meaning of an increase/decrease in the scores if this is not explained in the methods.
- Statistics: in which sense do you state that “the Mercuri scale was higher than baseline” if you did not run any statistics? I understand that the numeric value of the scale could be higher at follow-up if compared to baseline, but to me it is not acceptable to just claim “the score was higher”: was this significant? In my view, potentially, a slight increase in the score value could or could not mean that a real change in muscle quality or functionality has occurred: in this scenario, statistics and correlations may help to assess whether a certain observed alteration has a clinical/functional relevance.
This is then true for all the work: there is no comparison run between baseline and follow-up, which could be very easily assessed by comparing the scores at baseline and follow-up, by using a simple t-test if the normality of the dataset is confirmed.
By running a statistics between the MRI and PUL scores at baseline and follow-up, it would be possible to check which districts/muscles and functions are more compromised in a confident way; otherwise the work is just observational, on a small sample size, and thus it is difficult to appreciate its contribution (which I actually really see, but should be strengthened by running statistics).
- Results: table 1 and 2. I understand that you need to show a lot of data; however, these tables do not read well as they are very-small written and I had difficulties in the interpretation: maybe you could split these tables in more than 1, increase the font size and thus also simplify the captions which are very long and complex to read?
- Figures: why do you present in the first column the follow-up and in the second the baseline (blue and orange, respectively)? To my understanding, it would be more logic to present in the first column the baseline value, and in the second the follow-up.
- Discussion: “The possibility to explore all the domains allowed to define which muscles and domains are at higher risk of becoming progressively involved with increased age” (L291-292): again, if you want to state so, then statistics should support your findings.
L299-300: muscle “which”, not “who” – please amend.
“There was no constant association between baseline MR scores and follow up PUL scores but at shoulder level the highest risk of decreased function on PUL over a year was associated with moderate impairment on the baseline MRI scores (between 16 and 34)”, again, was the “association” statistically significant?
Author Response
The present work investigates the MRI and PUL variations over 1 year of observation in a sample of 27 DMD patients, in order to assess the feasibility and reliability of an integrated muscle imaging and functional testing in monitoring disease output and progression. The study is very interesting, as the relevance of the findings could be of great help in the clinical routine and practice when treating DMD patients; thus, I thank the author for their efforts.
We thank the reviewer for her/his comment
However, there are few points which I believe could be addressed in order to strengthen the results of this work, and also some suggestions to potentially help the paper becoming more reader-friendly.
- Methods: exclusion criteria. I understand you had to exclude those patients where MRI could not be collected for many reasons (cited in the paper). However, I was wondering whether this could represent a limitation of this work, as the proposed method for measuring muscle MRI and PUL cannot be applied to all patients. I understand there is now way to solve this, but the authors could cite it as a limitation of the study.
We thank the reviewer for the comment. We added a sentence in the discussion to ackowledge the limitation, as suggested.
Another limitation was that related to the exclusion criteria as MRI cannot be easily perfomed in patients with severe behavioural or cognitive problems or severe contractures.
- Methods: MRI scoring, L110-116: I don’t understand how you defined the score. Was it the sum of the scores of all the muscles analysed for each district (shoulder, arm and forearm)?
I.e. Did you sum the individual score of, for example, biceps brachii, brachialis and triceps brachii to obtain the score of the arm?
We thank the reviewer for the comment that gives us the possibility to explain the score in more detail. For each segment the sum of the scores of all the muscles analysed has been considered. This has been added.
For each segment the sum of the scores of all the muscles analysed has been considered: at shoulder level total shoulder scores were considered as mild if <16, moderate if between 16-34 and severe when > 34. At arm level total arm scores were considered as mild if < 4, moderate if between 4-7.5, severe when >7.5. At forearm total forearm scores were considered as mild if ≤ 6, moderate if between 6.5-36, severe when >36.
L199: PUL: I see there are references, however I would require the authors to briefly explain (in a more comprehensive manner) the PUL determination modality. When I read the paper indicated as ref. 8, it becomes clearer what the PUL 2.0 is; of course the authors cannot report the whole questionnaire, but rephrasing the chapter in a clearer way would certainly help an inexpert reader. For example, briefly explaining what the entry item is, and providing few examples of the other items (i.e. hand to mouth, hands lap to table) and how the score is defined (ex. 2: able; 1: with compensation; 0: unable). This would be particularly useful also at the light of the fact that all the results are presented referring to the increased/decreased PUL scores, and the reader has no idea of the functional meaning of an increase/decrease in the scores if this is not explained in the methods.
We thank the reviewer for the comment. This information has been added and the whole proforma has been added as supplementary figure.
The 22 items include item 1: Shoulder abduction both arms above head; item 2: Shoulder abduction to shoulder height; items 3 (no weights) and 4 (500g): Shoulder flexion to shoulder height; items 5 (500g) and 6 (1kg): Shoulder flexion above shoulder height; item 7: Hands to mouth; item 8: Hands lap to table; items 9, 10, and 11: Move weight on table (100g, 500g, 1kg); item 12: Lift heavy can diagonally-timed; item 13 (three cans) and 14 (five cans). No time; item 15: Remove lid; item 16: Tearing paper; item 17: Tracing path; item 18: Push on light; item 19: Supination; item 20: Pick up coins; item 21: Number diagram; item 22: Pick up 10g finger pinch.
- Statistics: in which sense do you state that “the Mercuri scale was higher than baseline” if you did not run any statistics? I understand that the numeric value of the scale could be higher at follow-up if compared to baseline, but to me it is not acceptable to just claim “the score was higher”: was this significant? In my view, potentially, a slight increase in the score value could or could not mean that a real change in muscle quality or functionality has occurred: in this scenario, statistics and correlations may help to assess whether a certain observed alteration has a clinical/functional relevance.
This is then true for all the work: there is no comparison run between baseline and follow-up, which could be very easily assessed by comparing the scores at baseline and follow-up, by using a simple t-test if the normality of the dataset is confirmed.
By running a statistics between the MRI and PUL scores at baseline and follow-up, it would be possible to check which districts/muscles and functions are more compromised in a confident way; otherwise the work is just observational, on a small sample size, and thus it is difficult to appreciate its contribution (which I actually really see, but should be strengthened by running statistics).
As suggested we perfomed the analysis between baseline and follow up on PUL and MRI .
There were some differences in total scores between baseline and follow up in each segment on the PUL indicating more functional impairment but this did not reach significance.
There were some differences in total scores between baseline and follow up in each segment indicating more muscle involvement on MRI but this did not reach significance.
This information has been added in the test
- Results: table 1 and 2. I understand that you need to show a lot of data; however, these tables do not read well as they are very-small written and I had difficulties in the interpretation: maybe you could split these tables in more than 1, increase the font size and thus also simplify the captions which are very long and complex to read?
Individual detailed data is often requested and we feel that this should be kept. When transferred to a pdf as requested by the instructionssto authors the table is less readible but the printed and online version will show the origibal table that can be further adjusted at proof level to make it more readible.
Anyway the table has been amended.
- Figures: why do you present in the first column the follow-up and in the second the baseline (blue and orange, respectively)? To my understanding, it would be more logic to present in the first column the baseline value, and in the second the follow-up.
We thank the reviewer, this has been amended.
- Discussion: “The possibility to explore all the domains allowed to define which muscles and domains are at higher risk of becoming progressively involved with increased age” (L291-292): again, if you want to state so, then statistics should support your findings.
There is an obvious increase in impairment even if does not reach significance, and this is in line with the well known progression in DMD.
L299-300: muscle “which”, not “who” – please amend.
This has been amended
“There was no constant association between baseline MR scores and follow up PUL scores but at shoulder level the highest risk of decreased function on PUL over a year was associated with moderate impairment on the baseline MRI scores (between 16 and 34)”, again, was the “association” statistically significant?
As suggestyed we run a statistycal correlation:
No significant differences were found at baseline and at follow up for both MRI and PUL in total scores and subtotal scores at each level (shoulder, arm and forearm). No significant correlation was found between baseline MRI total scores, baseline MRI subtotal scores of each level and PUL scores at follow up but for the shoulder level where a moderate correlation (r=0.41, p=0.01) was found between MRI shoulder scores at baseline (between 16 and 34), and PUL shoulder scores at follow up.
Reviewer 2 Report
The subject of the article is important, once Duchenne muscular dystrophy a severe type of muscular dystrophy that primarily affects boys. However, there are several flaws in the article that should be corrected
General comments:
- In the Abstract is good. However, there are several undefined abbreviations (e.g. MRI, DMP)
- The introduction lacks the relevance of this pathology, and manly the definition and the symptoms of this pathology.
- The table 1 are too small, and is not visible.
- The limitations of the study should be included.
Author Response
The subject of the article is important, once Duchenne muscular dystrophy a severe type of muscular dystrophy that primarily affects boys. However, there are several flaws in the article that should be corrected
We thank the reviewer for her/his comment
General comments:
- In the Abstract is good. However, there are several undefined abbreviations (e.g. MRI, DMP)
This has been amended
- The introduction lacks the relevance of this pathology, and manly the definition and the symptoms of this pathology.
We thank the reviewer for the observation. This has been added.
- The table 1 are too small, and is not visible.
See above
The limitations of the study should be included.
These have been added
Round 2
Reviewer 1 Report
I have read the new version of the paper, and the replies to my comments; I feel like the authors did a good job in revising their manuscript, which, for me, is now ready for publication.
Reviewer 2 Report
This article was revised appropriately.
I recommend accept